# Ezrin Peptide Therapy from HIV to COVID: Inhibition of Inflammation and Amplification of Adaptive Anti-Viral Immunity

**DOI:** 10.3390/ijms222111688

**Published:** 2021-10-28

**Authors:** Rupert D. Holms, Ravshan I. Ataullakhanov

**Affiliations:** 1Newal R&D Ltd., London NW1 7SX, UK; 2Institute of Immunology, Ministry of Health of the Russian Federation, 115478 Moscow, Russia; ravshan.ataullakhanov@gmail.com

**Keywords:** COVID, ezrin peptides, therapy

## Abstract

Human Ezrin Peptides (HEPs) are inhibitors of expression of IL-6 and other inflammatory cytokines, amplifiers of adaptive B cell and T cell immunity and enhancers of tissue repair. The mutation stable *C*-terminus of HIV gp120, mimics 69% of the “Hep-receptor”, a zipped α-helical structure in the middle of the α domain of human ezrin protein. Synthetic peptides homologous to the Hep-receptor of ezrin of five to fourteen amino acids, activate anti-viral immunity against a wide range of viruses (HIV, HCV, herpes, HPV, influenza and other human respiratory viruses). Human Ezrin Peptide One (HEP1) TEKKRRETVEREKE (brand name Gepon, registered for human use in Russia from 2001) is a successful treatment for opportunistic infections in HIV-infected patients. That treats HEP1and prevents mucosal candidiasis, herpes zoster outbreaks and infection-induced chronic diarrhea. There are clinical publications in Russian on the successful treatments of chronic recurrent vaginal candidiasis, acute and chronic enterocolitis and dysbacteriosis, which are accompanied by normalization of the mucosal microbiome, and the decline or disappearance of inflammation. HEP1 is also an effective treatment and prevention for recurrent inflammation and ulceration in the stomach, duodenum and colon. HEP1 and RepG3 GEKKRRETVEREGG (a derivative of HEP1) have been used successfully as an inhaled spray peptide solution to treat a small number of human volunteers with mild-to-moderate COVID, resulting from SARS-CoV-2 infection, based on earlier successes in treating acute viral respiratory disease with inflammatory complications. Ezrin peptides seem to correct a dysregulation of innate immune responses to SARS-CoV-2. They are also adjuvants of B cell adaptive immunity and increase antibody titres, resulting in protection from lethal virus infection of mice. In a clinical study in Moscow, orally administered HEP1 was shown to enhance antibody-titres produced in response to hepatitis-B vaccination. These very preliminary but promising results with ezrin peptide treatment of COVID must be replicated in large-scale randomised placebo controlled clinical studies, to be verified.

## 1. Introduction

### 1.1. Treatment of COVID and Prevention of SARS-CoV-2 Reinfection

There is a great need for a safe, effective, cheap and reliable therapy for acute and chronic COVID disease induced by SARS-CoV-2 infection. It is well established that high levels of inflammation and IL-6 expression in response to SARS-CoV-2 infection predicts severe COVID-19 and death [1,2,3]. It has also been demonstrated that 2019 SARS-CoV-2 spike protein subunit 1 (CoV2-S1) induces high levels of NF-κB activation, production of pro-inflammatory cytokines and epithelial damage in human bronchial epithelial cells [BCi-NS1.1 cell line] [2]. SARS-CoV-2 infection or spike protein expression in human epithelial cells [Huh7.5 and A549 cell lines] inhibits ACE2 expression leading to the induction of AT1 signalling, and promotion of IL-6/soluble IL-6R release [4]. GU-rich RNA from the SARS-CoV-2 spike protein can induce a TLR8-dependent pro-inflammatory cytokine response from human macrophages [5]. However, anti-IL-6 monoclonal antibody-blockade has not been a reliable therapy for COVID [6,7,8,9]. The ideal therapy should correct the dysfunction of the innate immune response to SARS-CoV-2 which leads to excess inflammation and over-expression of IL-6, as well as enhancing adaptive immune responses to resolve the infection and prevent re-infection [10,11]. Ezrin peptides have been tested in human volunteers for the treatment of mild-to-moderate COVID, and seem to have the potential to achieve these clinical endpoints [12,13,14].

Ezrin peptide therapy was reverse-engineered from HIV as a result of a discovery in 1993, during research into a possible peptide vaccine for HIV and AIDS [15]. An unexpected homology between the mutation-stable carboxy *C*-terminus of HIV gp120, and the central α domain of human ezrin, a multi-functional membrane-associated human protein, was found (Figure 1). Ezrin also regulates the original SARS coronavirus spike-protein interaction with cells and active ezrin inhibits SARS entry into cells [16].Both the *C*-terminus of HIV gp120 and the central α domain of ezrin are α-helix forming regions and are highly charged. The match of the amino-acid side chain charges and secondary structure suggested the HIV gp120 and human ezrin sequences have related functions [17,18,19,20,21].

### 1.2. Human Ezrin Peptides

Ezrin and its homologues radixin and moesin (the ERM proteins) are cell membrane- associated proteins with multiple binding sites for cell surface receptors, intracellular kinases and actin, forming protein complexes involved in cell-signalling, cell-shape and cell-motility. The 14 amino acid synthetic peptide TEKKRRETVEREKE known as human ezrin peptide one (HEP1), is the active pharmaceutical ingredient of Gepon, registered for human use in the Russian Federation since 2001 [22,23,24]. Gepon was registered as an immune-modulator for human topical and internal use to restore effective immunity in the treatment and prevention of viral, bacterial and fungal infections. Gepon was particularly effective at controlling opportunistic infections in HIV patients [25].

Ezrin peptides have been in clinical use in Russia for over twenty years and have an excellent safety record. HEP1 mimics part of a double α-helical zip-like structure called the Hep-receptor located at amino acids 308–372, in the central Alpha domain of human ezrin. Since 2001, the biological activity of Gepon TEKKRRETVEREKE has been extensively investigated in Russia. Successful clinical trials using Gepon for the treatment of acute viral respiratory infection (AVRI) with complications including pneumonia, were performed in Moscow over the previous eighteen years, in which the reduction of inflammation and IL-6 was achieved [26,27,28,29,30].

Ezrin peptides have immune amplification and anti-inflammatory properties. They are effective against HCV [31,32,33]. They are an effective treatment of a wide spectrum of other viruses such as herpes 1 and 2 and HPV [34,35]. Ezrin peptides are very effective at treating the infection and inflammation associated with *Candida* and other infections of the mucus membranes [36,37,38]. The anti-inflammatory properties and tissue regeneration properties of ezrin peptides have been applied to the treatment of ulcers of the gut, from oral ulcers, to stomach and duodenal ulcers, to lower gut ulceration associated with inflammatory bowel disease, trophic ulcers of the skin and radiation ulcers [39,40,41,42,43,44,45]. Ezrin peptides stimulate fibroblasts to repair and regenerate damaged tissues [46].

Gepon has been demonstrated to be an amplifier of immune responses, increasing anti-body titres by 2×× to 3×, while significantly inhibiting the expression of inflammatory cytokines: particularly IL-6, IL-1β, IL-8 and TNFα [47,48,49]. Regulatory Ezrin peptide glycine 3 (RepG3) is a 14 amino acid synthetic peptide, with 11 amino acids identical to Gepon, but with three terminal amino acids replaced with glycine (Figure 2). The modification of this ezrin peptide doubles the anti-inflammatory activity of RepG3 2× over Gepon, in a mouse dextran sulphate solution (DSS) gut inflammation model that mimics inflammatory bowel disease (IBD).

HEP1 is a 14 amino acid synthetic peptide mimicking part of the α domain of human ezrin known as the Hep-receptor, which is formed by amino acid 324 and amino acid 337 inclusive. The Hep-receptor is a zip-like structure comprising of two anti-parallel alpha-helices carrying alternating positively and negatively charged side chains. Ezrin peptides mimicking parts of the Hep-receptor are all biologically active.

### 1.3. Ezrin and the B-Cell Receptor (BCR)

Ezrin peptides suppress inflammation and amplify B cell responses, an ideal combination of characteristics to treat and prevent COVID induced by SARS-CoV-2. Considering the antibody-enhancing activity of ezrin peptides, it is interesting to consider the role of human cell-membrane-protein ezrin in antigen recognition by B-cells and BCR-signalling.

Ezrin and its homologues radixin and moesin, not only regulate the B-cell immune response, but they are also critical for leukocyte polarization, migration and intercellular adhesion [50,51]. Following engagement of the B-cell receptor (BCR) with antigen, B-cells contribute to the adaptive immune response to viral infection by transforming into antibody-producing plasma cells and long-lived memory cells. Ezrin is directly involved in cellular responsiveness and transmembrane signalling in B cells [52]. Ezrin and actin forms a protein complex with the BCR and coordinates the early events of B-cell activation to antigens [53]. Ezrin-actin complexes control BCR micro-clustering on the B-cell surface in response to antigenic stimulation [54]. The binding of antigen to the B cell Receptor results in the phosphorylation of tyrosine-353 located in the ezrin Hep-receptor, leading to JNK activation [55]. Variations of ezrin anchoring of the actin cytoskeleton to the cell membrane, controls the initiation and regulation of BCR activation [56]. 

## 2. Clinical Trials of Ezrin Peptide Treatment of Acute Viral Respiratory Infection (AVRI)

135 adults presenting acute viral respiratory infection (AVRI), 28% with pneumonia, were recruited at Department of Infectious Diseases, Moscow Hospital No 1, in collaboration with the Russian Ministry of Health Institute of Immunology, during 2008. Of the 135 AVRI patients, 49% of the cases were due to influenza, 13% to parainfluenza, 14% to adenovirus and 8% to respiratory syncytial virus (RSV), while 10% was due to mixed virus and 6% to unidentified virus. The patients volunteered for a clinical trial of solution-vapour treatment with ezrin-peptide TEKKRRETVEREKE (Gepon) administered to the airways [29].

All patients received standard symptomatic therapy: anti-inflammatory paracetamol, antihistamines, expectorants and inhalation of vaporized 0.2% sodium bicarbonate solution. Some patients with AVRI presented evidence of serious inflammation of the sinuses, bronchitis, obstruction of the pharynx, together with hyperaemia of the mucous membrane of the pharynx, swollen tonsils, sores on pharynx wall, and purulent deposits. Some patients complained of severe congestion, mucopurulent discharge from the nose and debilitating headache, which required X-ray examination of the nose. 

IgM, IgA, IgG and IgE were measured together with concentrations of *C*-reactive protein in the blood. Bacteriological analysis of the sputum was used to identify Mycobacterium tuberculosis (if present), non-specific micro-flora and any antibiotic drug-resistance.

Flow cytometry was used to count peripheral blood immune cell subpopulations, their activation markers and functional subtypes such as CD4+ T helper cells, CD8+ cytotoxic T lymphocytes and NK cells. Chemo-luminescence was applied for the functional study of granulocyte ex vivo response to zymosan, a fungal glucan recognized by TLR2 receptors. 

A sub-group of 48 adult patients presenting AVRI only (no pneumonia), were enrolled for a randomised clinical study of Gepon inhalation therapy and 26 patients were randomly assigned to the Gepon treatment group A and 22 to the Control Group A. Gepon treatment Group A, received 1 mg in 5 mL Ezrin-peptide TEKKRRETVEREKE (Gepon) solution vapour inhalation per treatment. Gepon was prepared for treatment in batches by dissolving 2 mg lyophilised Gepon in 10 mL of isotonic NaCl solution, resulting in a 0.02% Gepon solution. 5 mL of solution was added to an ultrasonic Beron inhaler and blown into the nasal cavity and airways of the patient, once a day, for 5 consecutive days. The total course of therapy was 5 mg of peptide. Ezrin peptide TEKKRRETVEREKE (Gepon) significantly accelerated recovery from of acute viral respiratory infection (AVRI) from over four days to two days

A sub-group of 49 patients had AVRI + throat inflammation, sinusitis, tonsillitis and bronchitis, all received standard therapy. Twelve patients received Gepon inhalation therapy in addition to standard symptomatic treatments. By the third day of hospitalisation, all 37 control group patients, suffered worsening symptoms and had to receive antibiotics for 5 to 7 days. In contrast, the 12 patients of Gepon Group B (*n* = 12) steadily improved and the duration of illness was significantly less. There was a rapid reduction of inflammation in patients who received Gepon inhalation. Bronchitis, laryngitis and sinusitis persisted for about 8 days in the Control Group B, while all 12 patients had recovered in Gepon Group B after 5 to 6 days.

Thirty eight patients suffered acute viral respiratory disease (AVRI) complicated with pneumonia. Patients suffered fever up to 39 °C and 82% presented a dry cough. The patients were randomised, and two subgroups were created: 20 patients were allocated to a control group and 18 patients were allocated to the Gepon treatment group and received 1 mg in 5 mL Gepon solution vapour inhalation per treatment. The total course of therapy was 5 mg of peptide.

Ezrin peptide TEKKRRETVEREKE (Gepon) significantly shortened the duration of fever, intoxication, headache and weakness, and accelerated recovery from acute viral respiratory infection (AVRI) complicated with pneumonia (Figure 3) [23]. 

Gepon was particularly effective at reversing high fever temperatures (39 °C), triggered by lung infection and inflammatory pneumonia. Gepon reduced by twenty per cent, the duration of shortness of breath, hypoxemia (low arterial blood gas), and the need for supplemental oxygen and breathing support.

On the first day of hospitalisation, there were 79 patients who presented ARVI without bacterial complications. Antibiotics had to be prescribed to 26 patients who received standard therapy only. In contrast there were only five cases of bacterial co-infection with patients who were receiving Gepon, who also needed antibiotic therapy. Gepon inhalation therapy had reduced the risk of bacterial infection in hospital by more than three times.

## 3. Clinical Trials of Ezrin Peptide Treatment of AVRI with Inflammation in Children

In year 2000, a clinical study into the efficacy intra-nasal ezrin-peptide-one (HEP1) TEKKRRETVEREKE (Gepon) solution treatment of acute viral respiratory infection [AVRI], inflammatory laryngeal-tracheal-bronchitis plus stenosis [SLTB] and recurrent croup [RC] was performed in 100 child patients, at the Morozov Moscow Children’s Hospital, in collaboration with The Russian Government Medical University, Moscow [25,30,57,58].

An assessed patient population of 125 children were offered immune status analysis, which was then compared to the average status of healthy children. Generally, a profound non-specific inflammatory response was being induced and maintained by the viral infection. In contrast, specific anti-viral immunity had been disrupted. The most consistent differences between the assessed child-patients and healthy children, were the large significant increases in the concentrations of inflammatory cytokines: IL-1β, IL-6, IL-8 and TNFα. Interleukin-6 (IL-6) was significantly elevated 2.1× in vivo and 9× in vitro, and hyper-elevated 26× when induced in vitro. IL-8 and TNFα also showed a similar pattern of hyper-elevation. 

One hindred children with AVRI were selected for the clinical study. The 50 children randomly allocated for the control group received standard symptomatic treatment only for viral inflammation of the airways. 13 children of the control group also received antibiotics for secondary bacterial infections. 

The 50 children randomly selected for the Gepon group received standard symptomatic treatment for viral inflammation of the airways plus Gepon therapy: 2 mg sterile lyophilised ezrin-peptide-one (HEP1) TEKKRRETVEREKE (brand name Gepon, produced by LLC Immapharma, Moscow, Russia) dissolved in 2 mL water to make a solution of 1 mg/mL. Gepon solution was delivered intra-nasally as 5 drops in each nasal passage, twice a day, for a period of 5 days (total administration of 2 mg Gepon in 2 mL). In the case of 7 children, who had AVRI with bacterial complications, antibiotic treatment was also applied in parallel to Gepon treatment (Figure 4).

No child-patient displayed any adverse reaction, nor any adverse drug interaction, when receiving Gepon. No side effects were observed. No allergic reactions were detected with Gepon therapy. No child-patient suffering from atopic dermatitis displayed any aggravation of illness. There were no hypersensitivities resulting from the intra-nasal introduction of Gepon and no evidence of any contra-indications at any patient age. All child-patients who received human-ezrin-peptide-one (HEP1).

TEKKRRETVEREKE (Gepon), in addition to standard symptomatic treatment, benefitted from a significant shortening in the duration of the clinical symptoms, which was independent of the severity of AVRI or SLTB. The child-patients experienced the rapid decrease in the duration of symptoms of disease, regardless of the severity of the acute viral respiratory infection (ARVI), the amount of inflammation of the airways, stenosis of the larynx or upper-airway obstruction. The most remarkable effect of human-ezrin-peptide therapy was the rapid and complete elimination of conjunctivitis of the eye.

Comparisons between the 50 child-patient Gepon group and 50 child-patient control group, showed that the duration of the fever and other manifestations of intoxication syndrome, including malaise, reduced appetite, weakness, sleepiness, and decrease in physical activity, were all reduced. 

All child-patients presented with dry-cough and fever, but after they received Gepon therapy, they benefitted from a reduction in the duration of fever by 3.2× times to only two days, and the duration of dry cough by 1.8×, so that it stopped in less than four days. Duration of rhinitis was 2.2× less and laryngitis was reduced by 2.3×. Croup-cough disappeared on Day-3 of treatment with Gepon. In contrast, only 38% of the control group managed to eliminate croup-cough in the same period. As a result of Gepon therapy, dissolution of mucus and appearance of productive wet cough, a sign of recovery occurred on Day-2 of Gepon therapy. In the control group, productive wet-cough only got established after Day-5. Regardless of the degree of inflammation and laryngeal-tracheal-bronchitis with stenosis (SLTB) in the Gepon group, 67% of cases had recovered by Day-2 of treatment. In the same two days, 72% of the children in the Gepon group increased sputum density, while only 46% of the control group improved. 

Gepon treatment reduced the bacterial complications requiring antibiotics. Gepon even reduced the manifestations of atopic dermatitis in the child patients. In the child-patients who were suffering more severe AVRI + SLTB, human-ezrin-peptide-one (HEP1) TEKKRRETVEREKE (Gepon) was demonstrated to be a rapid acting treatment. Fever was shut down after 30 h of Gepon therapy, whereas the control AVRI+SLTB subgroup continued to suffer fever for 3 days. Maximum clinical benefit with Gepon was achieved in 67% of cases, within 48 h of receiving the treatment. 

Dry-cough disappeared in less than 60 h in three-quarters of the child-patients in the Gepon AVRI+SLTB subgroup, whereas only about a third of child-patients recovered from dry-cough in the control AVRI + SLTB subgroup, over the same period. In the first two days of therapy, 72% of the child-patients in the Gepon AVRI + SLTB subgroup benef C ited from the alteration of the mucus consistency towards dissolution. In contrast, in the control AVRI + SLTB sub-group, only 46% of the child-patients enjoyed this improvement in symptoms. 

Gepon therapy significantly benefitted all the very sick child-patients who required antibiotics. In the sub-group of child-patients who were suffering from both AVRI and SLTB, and who had been prescribed antibiotics to manage secondary bacterial infection, Gepon therapy granted them a significant shortening of the clinical symptoms and reduction in the duration of the antibiotics therapy. In the sub-group of child-patients suffering from AVRI and SLTB, and who were treated with antibiotics: Gepon therapy eliminated conjunctivitis within hours, and reduced fever to under two days, compared to over four days in the control sub-group. Gepon more than halved the average duration of stenosis of the larynx to around 34 h. In addition, Gepon therapy given to child-patients suffering AVRI + SLTB who were on antibiotics, reduced the duration of wet-cough to three days, compared to ten days observed in the control group. 

Child-patients who had received Gepon therapy, had a significant decline in recurrence of respiratory disease during the 3-month follow-up period of observation after treatment (Figure 5). On the rarer occasions when disease did re-occur, the illness progressed in a much milder form, and for shorter duration. After the first Gepon treatment for AVRI, only mild 3-day episodes of disease recurred, if at all, and the child-patients did not need hospitalisation. Gepon eliminated secondary bacterial complications requiring antibiotic therapy in almost all child-patients. In the control group, there were no such reductions in severity of recurring disease were observed.

During the subsequent 3 months of observation in the Gepon group, the number of episodes of AVRI was 0.5 per patient, whereas in the control group it was 1.6 per patient. The duration of one-episode of AVRI in the Gepon group, was 3.2 ± 0.3 days, whereas in the control group it was 6.9 ± 0.1 days. In those child-patients who received Gepon therapy but then fell ill again in the following 3 months, the AVRI was very mild, the duration of fever was reduced 3.2×, the duration of rhinitis 2.1×, and productive cough appeared on the average, on Day-2, compared to Day-5 in the control group.

In the Gepon subgroup with AVRI only, prior to therapy there were 17 cases of AVRI registered in 3 months. However, during the 3 months of observation following Gepon therapy, only eight cases of AVRI were recorded. In contrast in the control group, prior to therapy there were 16 cases of AVRI registered in 3 months and during the 3 months of observation following therapy, there were 13 cases of AVRI. Prior Gepon therapy reduced the duration of fever from 2.5 days to 0.7 days, reduced duration of wet cough from 5.1 days to 2.2 days, the duration of AVRI episodes from 6.9 days to 3.2 days and eliminated secondary bacterial infections and the need for antibiotics. 

In child-patients with AVRI + SLTB who had received Gepon therapy, the frequency of respiratory diseases 3 months after completion of therapy, were reduced by 60%. Gepon reduced the frequency of respiratory disease in child-patients with AVRI + SLTB from 1.8 in the three months prior to therapy to 0.69 in the three months after completion of therapy. In child-patients with recurrent AVRI, who had received Gepon therapy, the frequency of respiratory diseases 3 months after completion of therapy were also reduced by 60%. Gepon reduced frequency of respiratory diseases in children with recurrent AVRI (but no SLTB), from 3.1 in the three months prior to therapy, to only 1.1 in the three months after completion of therapy.

After the successful completion of the clinical study with intra-nasal ezrin-peptide TEKKRRETVEREKE (Gepon) therapy, the prophylactic efficiency of Gepon was assessed in child-patients who regularly suffered from recurrent AVRI. Child-patients attending hospital were assessed for the frequency of recurrent AVRI, longevity of AVRI, the type of clinical symptoms (fever, intoxication, sputum production and rhinitis) and the associated increases in allergic reactions, duration and enlargement of swollen lymph nodes, inflammation of the pharynx and tonsils, and the development of obstructive bronchitis or croup syndrome. 

In the prophylaxis study, the prophylaxis treatment regime was 1 drop of Gepon solution (1 mg/mL) into each nasal passage, 3 times a day, for 4 weeks. The result was no AVRI cases being registered over the following three months in the Gepon group. In contrast in the control group 0.6 cases per child were registered. 

Human-ezrin-peptide-one (HEP1) TEKKRRETVEREKE (Gepon) restored order to the immune responses dysregulated by pathogenic respiratory viruses, while suppressing non-specific inflammation. Gepon inhibited the expression of the inflammatory cytokines IL-1, IL-6, IL-8 and TNFα triggered by viral replication, while at the same time stimulated tissue repair and recovery processes. Gepon stimulated fibroblasts to repair the disturbed epithelial barrier to restore effective protection to bacterial, fungal and viral infection of the mucus membranes lining the airways.

It is remarkable how such a complex disease process as acute viral respiratory infection (AVRI) which induces laryngeal-tracheal-bronchitis with stenosis (SLTB) and croup, was so gently but effectively reversed by Gepon therapy. In children with AVRI + SLTB, simple intra-nasal therapy with TEKKRRETVEREKE solution, reliably reduced the duration and intensity of fever, reduced the concentrations of inflammatory cytokines, reduced the severity and duration of stenosis of the larynx, reduced the inflammation of the larynx, and reduced and terminated the period of dry cough, which was converted into productive-cough, co-incident with liquefaction of mucus in the nose and throat. Generally, Human-ezrin-peptide-one (HEP1) TEKKRRETVEREKE (Gepon) decreased the morbidity of AVRI by almost 3 times, reduced the duration and severity of recurrent AVRI, as well as reducing the annual incidents of AVRI.

In addition, long term benefits have been observed with ezrin-peptides. For three months after Gepon treatment, there was no recurrence of respiratory obstruction, no re-hospitalisation was required for normally chronic recurrent patients. Gepon also reduced the need for the treatment of secondary bacterial infection with antibiotics. Earlier studies demonstrated that ezrin-peptides could significantly enhance specific humoral immunity against infections, even in AIDS patients, amplifying antibody production against opportunistic infections [19]. Intra-nasal administration of Gepon is remarkably non-toxic and safe. No side effects nor unfavourable drug interactions were detected. There are no known contra-indications for Gepon for any age of patient. Thus, Gepon is a remarkably effective treatment for AVRI and SLTB croup. 

A model of chemically-induced acute colonic inflammation in mice, using oral administration of dextran sulphate solution (DSS), results in general inflammatory processes associated with weight loss and histopathology features mimicking some clinical features of inflammatory bowel disease (IBD). This model was used to compare the in vivo anti-inflammatory activity of HEP1 TEKKRRETVEREKE (Gepon) with RepG3 GEKKRRETVEREGG (V2). In this model, RepG3 was twice as effective as HEP1 (Gepon) at suppressing IL-6 expression, in experiments performed by Dr. Marina Chulkina (Figure 6) [59].

Human peripheral white blood cells stimulated with PHA express inflammatory cytokine mRNA to produce inflammatory cytokines. Ezrin peptide Rupe312, which contains the core sequence of RepG3, inhibits the expression of IL-6, IL-1β, IL-8 and TNFα but IL-2 expression is unaffected (Figure 7). A 95% reduction of IL-6 expression was observed with 100 ng/mL Rupe312 [35,60].

## 4. Human Ezrin Peptides Enhance Antibody Formation in Mice and Humans

Synthetic peptides mimicking the Hep-receptor of the α domain of human ezrin protein amplify antibody formation. The enhancement of antibody production correlates with protection from lethal infections in animal models. In laboratory mice, HEP1 (synthetic peptide TEKKRRETVEREKE) and other ezrin peptides have been demonstrated to enhance antibody formation. HEP1 increases the intensity of T cell-dependent immune responses to both soluble heterologous antigens such as egg albumin (EA) and cellular heterologous antigens such as sheep erythrocytes (SE).

HEP1 dissolved in 0.2 mL physiological saline was effective as an adjuvant injected intraperitoneally, in a dose range between 10 ng and 10 µg per mouse, either 60 min before administration of soluble EA antigen, or simultaneously with SE cellular antigen immunization. In addition, HEP1 was also an effective adjuvant when given orally, at a daily dose of 0.5 mg HEP1/0.2 mL water per mouse (25 mg/kg) for 20 days prior to immunization. Relative to control mice, the adjuvant activity of HEP1 caused 2.5× more IgM secreting plasma cells (differentiated mature B Cells) in the spleens, 2× more antigen-specific agglutinins and 3× more antigen-specific haemolysins, in the peripheral blood of treated mice. Primary and booster immunization with a mixture of HEP1 with soluble antigen, resulted in a 2× to 7× increase in IgG antigen-specific antibodies during the secondary immune response after antigen-booster [28,61,62,63,64].

In mice that had received oral pre-treatment with HEP1 for 20 days prior to immunisation, after secondary booster immunization with heterologous sheep erythrocytes (SE) cells, the titres of serum haemagglutinins were 2× higher, and the titres of serum haemolysin were 3× higher, than the control group of 27 mice who had received placebo-water only. An immune-adjuvant action was also observed with the intraperitoneal administration of HEP1 in the dose-range of 10 ng to 10 µg per mouse, with optimal dose from 0.1 μg to 1 μg of HEP1 per mouse (Figure 8).

Immuno-adjuvant activity of intraperitoneal injection of four different doses of HEP1 (ten-fold increases in concentrations from 10 ng to 10 µg per mouse), one hour prior to injection of 5 million heterologous sheep erythrocytes (SE), resulting in antibody-forming cells (ab.Cells) secreting SE-specific IgM antibodies in the spleens of mice, calculated as the number of antibody-forming cells in the spleen in HEP1 treated mice, divided by the number of antibody-forming cells in control spleens of mice immunized with SE cells only. 

## 5. Adjuvant Effect of HEP1 on the Production of IgM Antibodies to a Cellular Antigen

An adjuvant effect in mice was observed with an intraperitoneal injection of HEP1 solution on primary production of IgM antibodies to a cellular antigen. Mice received an intraperitoneal injection of 0.1 µg of HEP1, one hour before the injection of 5 × 10^6^ sheep erythrocytes (SE-cells). Four days after the intraperitoneal injection of SE-cells, the spleens of the mice had on average 1500 ± 200 antibody-forming cells secreting SE-specific IgM antibodies. In contrast, in the absence of HEP1 solution treatment, on average only 600 ± 75 antibody-forming cells secreting SE-specific IgM antibodies accumulated in the spleens of mice (50 antibody-forming cells were detected in the untreated control mice). 

In mice, an intraperitoneal injection of HEP1 (0.01 to 10 µg per mouse) one hour before immunisation with heterologous erythrocytes, is an effective adjuvant that can enhance the primary synthesis of IgM to antigens. Injection of HEP1 before or during exposure to heterologous antigens, on average increased the number of antigen-specific anti-body producing cells in the spleen by 2.5 times.

An immuno-adjuvant effect was also observed with a 20-day course of oral HEP1 solution pre-treatment. A group of 27 (CBA × C57Bl) F1 mice were given a 20-day course of oral 0.5 mg/0.2 mL HEP1 solution treatment, prior to sheep erythrocyte SE immunisation and booster, and compared to an equivalent control group of mice who received water placebo only. After the 20-day course of oral HEP1 solution treatment of 0.5 mg per day per mouse, an immunisation with 20 million SE cells followed by a booster of 20 million SE cells 21 days later, was performed. The serum haemagglutinin and haemolysin levels due to IgM in the peripheral blood were measured at weekly intervals (Figure 9).

The immune response to the foreign cellular antigen resulted in enhanced IgM antibody production in the oral HEP1 treated mice, compared to the control mice who did not receive HEP1. The adjuvant effect on serum haemagglutinin and haemolysin levels due to IgM in the peripheral blood, was clearly visible during the secondary immune response to the booster of SE cells. In mice, oral HEP1 solution pre-treatment, administered as a course of oral 0.5 mg/0.2 mL per day per mouse for twenty days prior to exposure to foreign antigens, increased the T-dependent antibody response to the antigens.

## 6. Immunoadjuvant Effect of Homologue Peptides of HEP1

The biological activity of fragments of HEP1 (see Figure 2) were investigated to determine the active regions of the peptide. Intra-peritoneal (i/p) immunisations of mice with five million sheep erythrocyte (SE) cells, led to immune responses to the foreign antigens four days later. Administration of HEP1 or homologue peptides of HEP1to mice, one hour before intraperitoneal immunisations with five million sheep erythrocyte (SE) cells, led to significant increases in the immune responses to the foreign antigens four days later. On average the spleens of mice accumulated 600 IgM antibody producing cells specific for SE cells (APCs). Generally, HEP1 and ezrin peptide fragments multiplied the number of APCs by 2× to 3×, but the dose-response pattern varied. Immuno-adjuvant activity of an intraperitoneal injection of four different doses of HEP1 and other ezrin peptide fragments (from 10 ng to 10 μg per mouse), immediately prior to a primary i/p immunisation injection of 5 million heterologous sheep erythrocytes (SE), resulted in ab.Cells secreting SE-specific IgM antibodies in the spleens of mice, calculated as the number of spleen ab.Cells in ezrin peptide treated mice divided by the number of control spleen ab.Cells in mice immunized with SE cells only (Figure 10).

## 7. Complex Adjuvant Effect of Ezrin Peptides with Egg-Albumin Antigen in Mice

HEP1 is an effective adjuvant in mice, which can enhance the secondary synthesis of IgG antibodies to soluble heterologous antigens. Injection of HEP1 at the same time as heterologous water-soluble egg-albumin led to an average of a 5× to 10× enhancement in titre of IgG antibodies in a secondary immune response after a booster immunisation. It was discovered that all the synthetic fragments of HEP1 between 5 to 12 amino acids in length that were tested, have a related immune-adjuvant activity, which suggests they are all acting on the stability of the Hep-receptor, and that the adjuvant activity may arise from conformational changes in ezrin, which leads to downstream signalling events. However there appear to be three distinct regions of activity in the 14 amino acid synthetic peptide HEP1: the *N*-terminal 5 amino acids, the core 10 amino acids and the *C*-terminal 5 amino acids. Sequence HP-10–14 EREKE seems to be responsible for the low concentration adjuvant activity-peak and sequence HP 1–5 TEKKR seems to be responsible for the high concentration adjuvant activity-peak (Figure 11 and Figure 12).

## 8. Immune Protection with Ezrin Peptides from Lethal Herpes Infection in Mice

An experimental model of lethal herpes virus infection of mice was developed in Russia to test the efficacy of immunostimulant molecules. In this lethal dose viral infection model, mice die within 48 h of an acute herpes simplex virus one (HSV-1 strain L2) infection. It was demonstrated that prior treatment with Ridostin, a mixture of double-stranded and single-stranded RNA from *Saccharomyces cerevisiae*, was a protective immunostimulant, allowing mice to survive more than four times longer than untreated controls. Ridostin went on to be tested in human clinical trials and was approved for human use in Russia [65]. 

Three ezrin Hep-receptor peptides: HEP1 and Rupe2032 mimicking the two helices forming the two sides of the Hep-receptor and Rupe1024, mimicking the MM hinge at the bottom of the Hep-receptor, were tested for efficacy in immune protection from lethal herpes-virus infection in mice (Figure 13) [66]. 

These Ezrin Hep-receptor peptides were tested for adjuvant activity at three different concentrations (1 or 10 or 100 ng per mouse) in the mouse lethal herpes virus infection model. Each of the nine ezrin peptide treatment groups, were a group of 20 laboratory B/P (10–12 g) white mice which were intraperitoneally injected with solutions of Hep-receptor peptides (1 or 10 or 100 ng/0.2 mL saline per mouse), 48 h and 24 h before being given a lethal intraperitoneal injection of herpes simplex virus (HSV-1 strain L2) at a titre of 1000 LD50 in 0.2 mL medium (titre 3.5 Log/0.2 mL).

The negative control (red bar) was a group of 20 laboratory B/P (10–12 g) white mice, which did not receive any adjuvant, before being given a lethal intraperitoneal injection of herpes virus (HSV-1 strain L2) at a titre of 1000 LD_50_ in 0.2 mL medium (titre 3.5 Log/0.2 mL). The positive control (purple bar) was a group of 20 laboratory B/P (10–12 g) white mice, which received an intraperitoneal injection of Ridostin (100 µg per mouse) 48 h and 24 h before being given a lethal intraperitoneal injection of herpes virus (HSV-1 strain L2) at a titre of 1000 LD50 in 0.2 mL medium (titre 3.5 Log/0.2 mL).

The mice were observed for 14 days. The untreated HSV-1 infected control group mice died in 1.25 days. The Ridostin pre-treatment group survived 6 days. The Ezrin peptide pre-treatment group survived between 5.4 and 10.1 days. It was discovered that only 0.1 µg (100 ng) of ezrin peptide per mouse, was more effective than one hundred micrograms (100,000 ng) of Ridostin per mouse, a one thousand times improvement in protection of the animals by ezrin peptides from lethal herpes HSV-1 infection. All the ezrin peptides had a related protective effect in the dose range between 1 ng per mouse up to 100 ng per mouse (Figure 14). 

Patients who complete a course of vaccination are considered immune to HBV, if their anti-HBs (hepatitis B surface antibody) titre is greater than 10 milli International Units per millilitre (mIU/mL). In a hepatitis B vaccination study in 185 children with malignant tumours, using Kombiotech (KHBV) vaccine for the prevention of hepatitis B infection with or without various adjuvants, a sub-group of 15 children were randomly selected for KHBV vaccination combined with oral HEP1 adjuvant (HEP1 group), and compared to 16 children who received KHBV vaccination without adjuvant (control group) [53] (Figure 15).

The course of KHBV vaccination was a series of three injections at monthly intervals followed by a fourth injection six months after the start of vaccination. In the HEP1 group, 2 mg/2 mL of HEP1 solution was given sub-lingually, held for two minutes in the mouth then swallowed before each of the four KHBV injections. In the control group (*n* = 16), 14/16 children (87%) seroconverted, with an average titre of 90.4 mIU/mL. In the HEP1 Group (*n* = 15), 13/15 children (87%) seroconverted, but with double the average titre of 181.1 mIU/mL. HEP1 also increased the per cent of patients who responded in the 100–1000 mIU/mL range by 2× and increased the per cent of patients who responded in the greater than 1000 mIU/mL range by 4×. 

## 9. Discussion

Our current hypothesis for the mechanism of action of ezrin peptides, is they act on a “receptor” conformation of human ezrin, in which the ezrin α domain is exposed on the external surface of the cell-membrane of lymphocytes, epithelial cells, fibroblasts, monocytes and macrophages. Mouse-monoclonal anti-ezrin antibody (6F1A9) (ab205381), raised to a recombinant fragment corresponding to human-ezrin amino-acids 292–464 containing the Hep-receptor and part of the α domain expressed in E. Coli, binds to human-ezrin exposed on the cell-surface of adherent epithelial luminal cell line MCF7 and HeLa cells (another epithelial cell line), demonstrated by flow cytometric analysis [67].

Highly charged ezrin peptides are believed to rapidly interact with the charged zip-like structure of the Hep-receptor in the α domain, inducing an allosteric effect that favours the opening of the submembrane active form of ezrin. (Figure 16). The active conformation of ezrin organises cell signalling complexes and amplifies down-stream signalling cascades. Cell signalling experiments in fibroblasts, NK cells and epithelial cell lines demonstrated rapid activation (in less than 10 min) of both the ras > raf > MEK > ERK pathway and also the PI3K > AKT pathway, showing ezrin peptides quickly engage a cell-surface receptor. In B-lymphocytes, activated ezrin forms complexes with the B cell receptor and signalling proteins, enhancing B cell activation and amplifying anti-body production.

The mechanism of ezrin peptide suppression of IL-6 expression is still speculative. Ezrin peptides trigger a signal down the ras > raf > MEK > ERK > RSK cascade, which may activate transcription factor CREB. IL-6 expression is mainly mediated by transcription factor NFκB, but active CREB competes for transcription co-factors CBP and p300 and inhibits NFκB mediated transcription (Figure 17) [68].

## Figures and Tables

**Figure 1 ijms-22-11688-f001:**
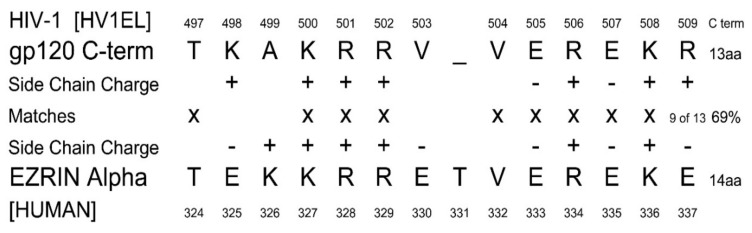
Homology of the mutation-stable *C*-terminus of HIV gp120 and the α domain of human ezrin.

**Figure 2 ijms-22-11688-f002:**
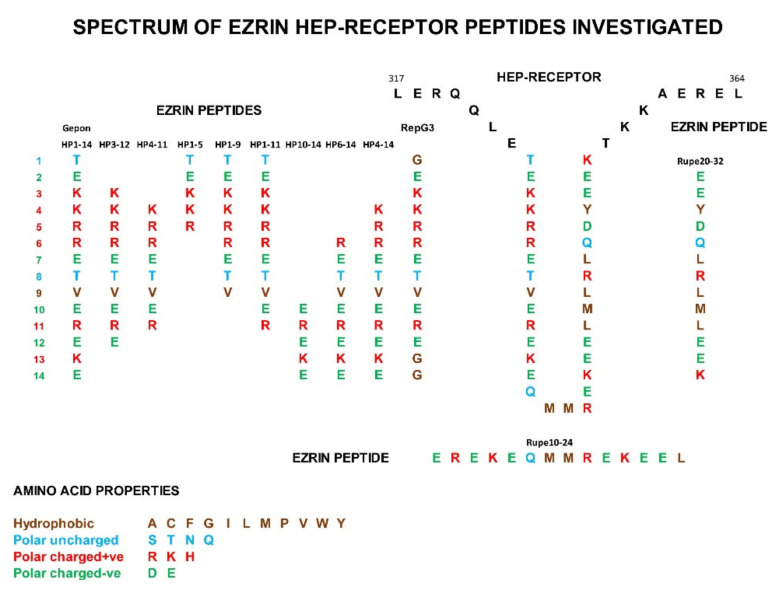
The electrostatically charged Hep-receptor in the α domain of human ezrin, comprises of two four-turn anti-parallel α-helices. Peptides from 5 to 14 amino acids in length that mimic parts of the Hep-receptor were synthesized and investigated for biological activity.

**Figure 3 ijms-22-11688-f003:**
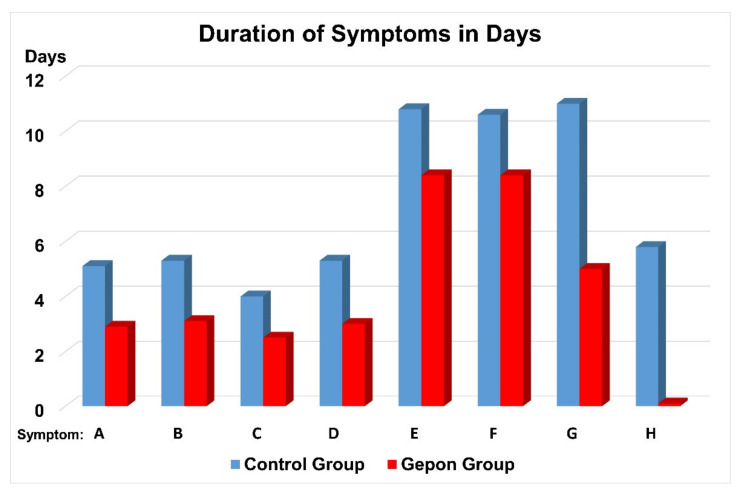
Duration of symptoms in days in patients with acute viral respiratory infection and pneumonia. Symptoms: A—fever, B—intoxication, C—headache, D—weakness, E—inflamed tonsils, F—mucosal hyperaemia, G—swollen lymphnode and H—purulent stomatitis. Control Group (blue), Gepon Group (red). The patients of the Gepon Group received 1 mg Gepon (TEKKRRETVEREKE) peptide dissolved in 5 mL isotonic NaCl solution (a 0.02% solution), blown into the nasal cavity and airways using an ultrasonic Beron inhaler, once a day for 5 consecutive days. The total course of therapy was 5 mg of Gepon.

**Figure 4 ijms-22-11688-f004:**
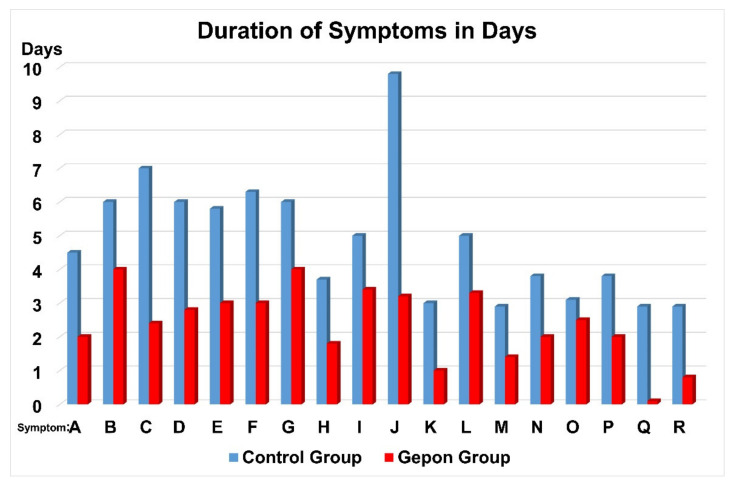
Duration of clinical symptoms (days). Control Group *n* = 50 (blue) Gepon group *n* = 50 (red). Horizontal divisions in days. Symptoms: A—Fever, B—difficulty in breathing, C—serous rhinitis, D—swelling of nasal mucous membrane, E—hyperaemia of tissues, F—pharyngitis, G—swollen palatine glands, H—hoarseness of voice, I—dry cough, J—moist cough, K—stenosis of larynx, L—swollen neck lymph nodes, M—reduction in the appetite, N—weakness, O—sleepiness, P—reduction in physical activity, Q—conjunctivitis, R—complications.

**Figure 5 ijms-22-11688-f005:**
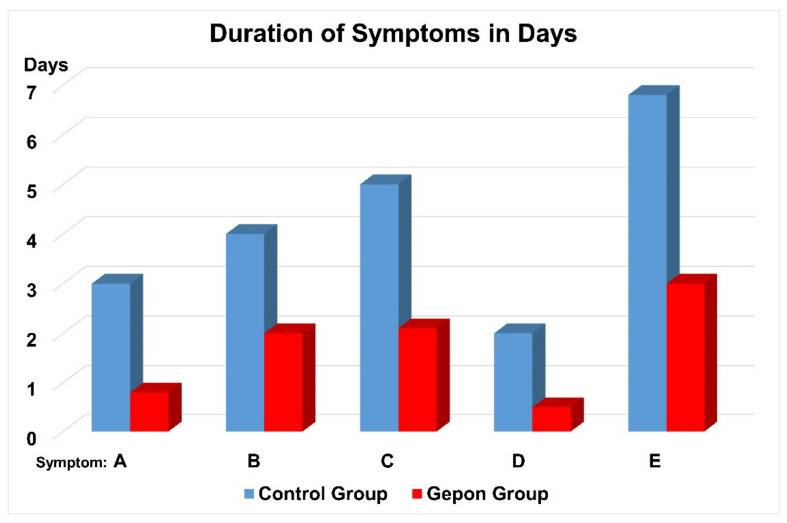
Three-month post-therapy follow-up. Control Group *n* = 50 (blue). Gepon group *n* = 50 (red). Duration of clinical symptoms in days (horizontal divisions). Symptom: A—Fever, B—Rhinitis, C—Wet-Cough, D—Antibiotics, E—AVRI. (drawn in Excel).

**Figure 6 ijms-22-11688-f006:**
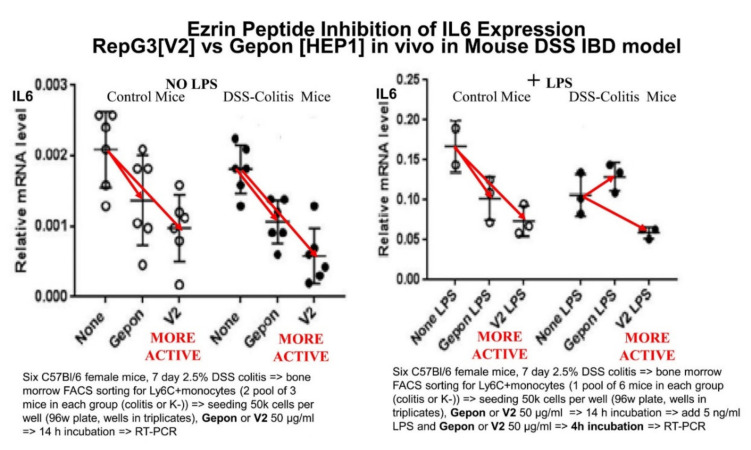
Ezrin peptides HEP1 TEKKRRETVEREKE (Gepon) and RepG3 GEKKRRETVEREGG (V2) have anti-inflammatory effects in vivo that correlated to reductions in IL-6 mRNA expression in both control mice (no DSS) and DSS-colitisinduced mice, in the presence and absence of LPS. The length of the red arrows draw attention to the relative amount of inhibition of IL-6 expression (LPS means lipopolysaccharide, a major component of gram-negative bacteria cell-walls that is a potent activator of monocytes/macrophages and causes an acute inflammatory response and release of inflammatory cytokines).

**Figure 7 ijms-22-11688-f007:**
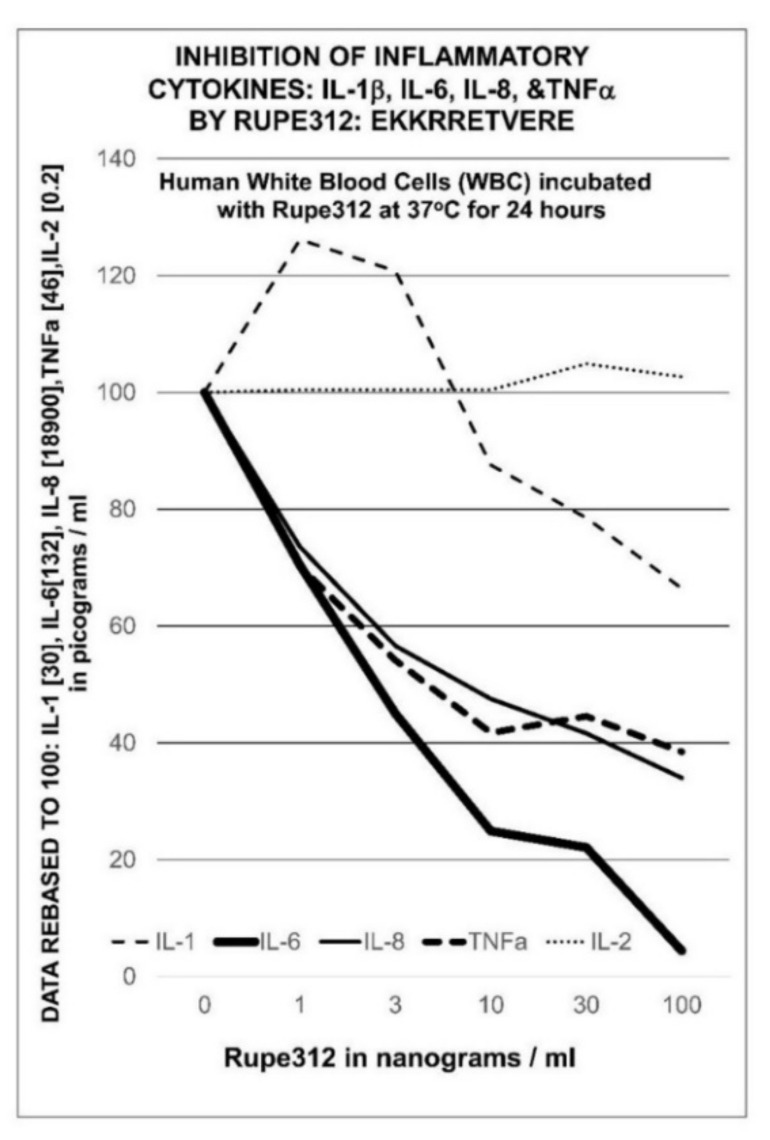
Inhibition of inflammatory cytokine expression in human WBC induced by 3 µg/mL PHA (phytohaemagglutinin) at 37 °C for 24 h, by ezrin peptide Rupe312 (the active sequence of RepG3) at concentrations of 1, 3, 10, 30, 100 nanograms per mL culture medium. Cytokine concentrations were measured in picograms per mL then expressed relative to the zero-Rupe312 control-value rebased to 100.

**Figure 8 ijms-22-11688-f008:**
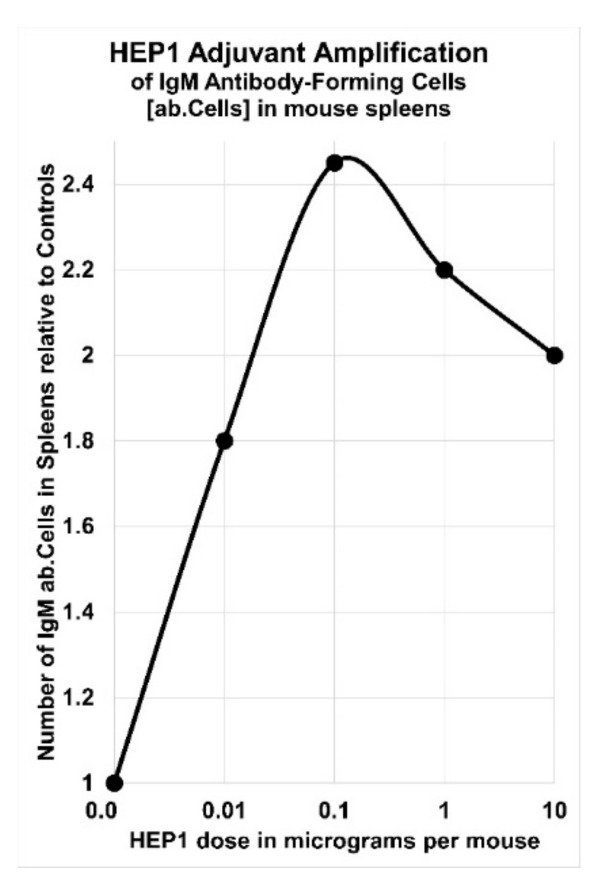
Immuno-adjuvant activity (measured as the number of IgM antibody-forming cells in spleens of mice, relative to control mice) as a result of intraperitoneal injection of 4 different doses of HEP1 (0.001, 0.01, 0.1, 1 and 10 µg per mouse) plotted in Excel.

**Figure 9 ijms-22-11688-f009:**
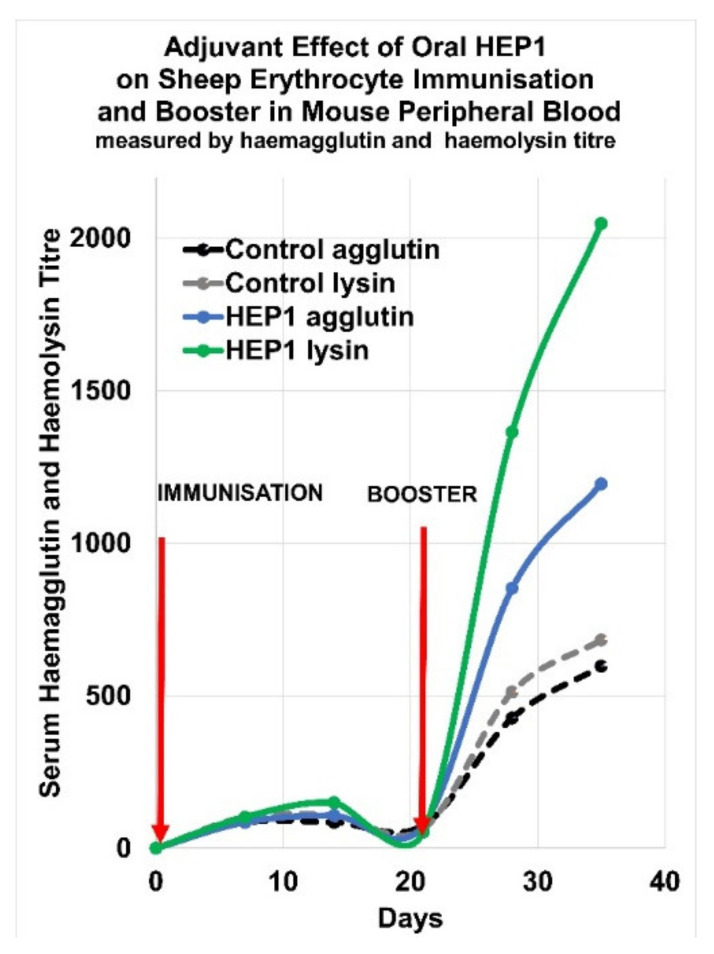
Adjuvant effect of oral HEP1 pre-treatment during 20 days prior to immunisation (Day 1) and booster (Day 21) immunisation with 20 million sheep erythrocyte (SE cells) in mice, measured by serum haemaglutin and haemolysin titre and compared to control mice with no oral HEP1 pre-treatment plotted in Excel.

**Figure 10 ijms-22-11688-f010:**
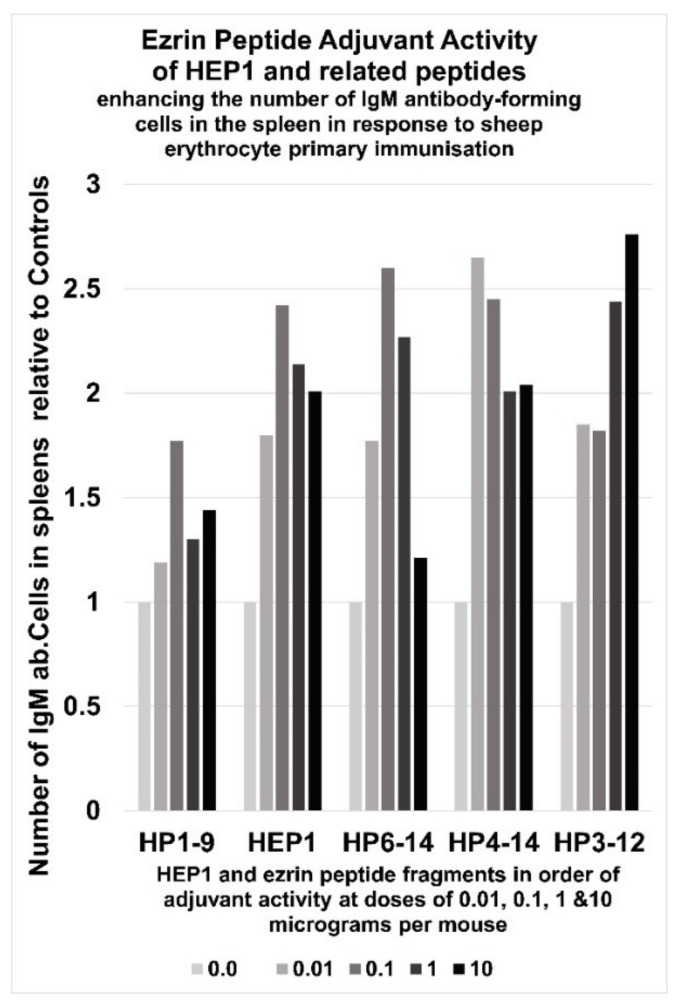
Ezrin peptide adjuvant activity of HEP1 and related peptides, enhances the number of IgM antibody-forming cells in the spleens of mice in response to sheep erythrocyte primary immunisation. HP1–9 TEKKRRETV, HEP1 TEKKRRETVEREKE, HP6–14 RETVEREKE, HP4–14 KRRETVEREKE and HP-3–12 KKRRETVERE.

**Figure 11 ijms-22-11688-f011:**
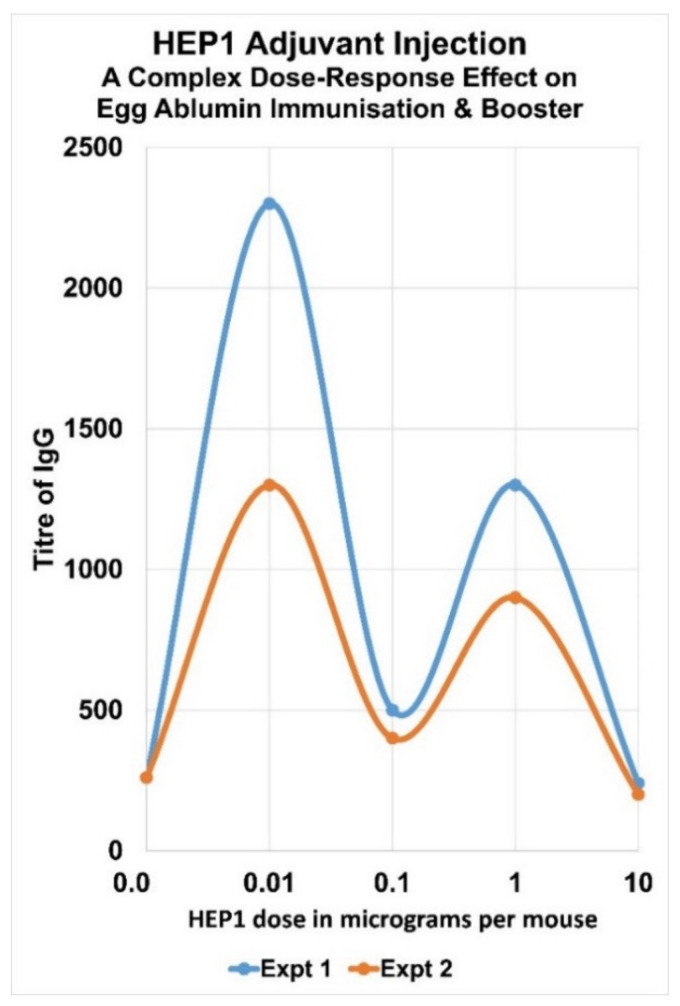
Two separate experiments demonstrated the complex dose-dependence of the adjuvant effect of HEP1 adminstered at 0.01, 0.1, 1 and 10 µg per mouse. This adjuvant activity of HEP1 was demonstrated in mice immunised and boosted with 50 µg of soluble heterologous egg albumin EA antigen simultaneously with HEP1, to produce serum IgG antibodies. The immune response 7 days after immunisation and booster with EA antigen, was expressed as ELISA IgG titre, plotted by Excel.

**Figure 12 ijms-22-11688-f012:**
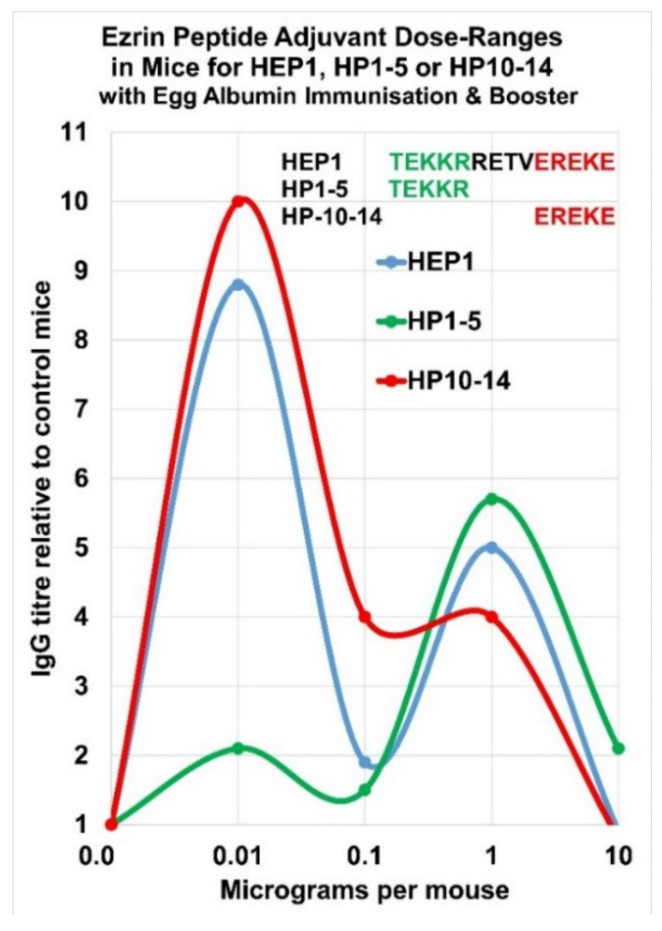
The two separate peaks of adjuvant activity were observed in mice i/p-injected with HEP1 or with ezrin peptide fragment HP1–5 TEKKR or with ezrin peptide fragment HP10–14 EREKE, and simultaneously immunised then boosted with 50 µg of soluble heterologous egg albumin EA antigen, to produce serum IgG antibodies. The immune response 7 days after immunisation and booster with EA antigen was expressed relative to the ELISA IgG titre in control mice, which had received injections of EA only, plotted by Excel.

**Figure 13 ijms-22-11688-f013:**
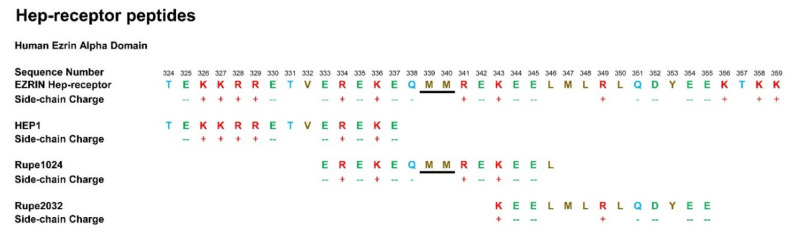
Over-lapping synthetic peptides mimicking the two helices and the MM hinge of the Hep-receptor of ezrin. The colours represent the physical amino acid properties: brown; hydrophobic, blue; polar uncharged, red; polar +ve charged, green polar –ve charged.

**Figure 14 ijms-22-11688-f014:**
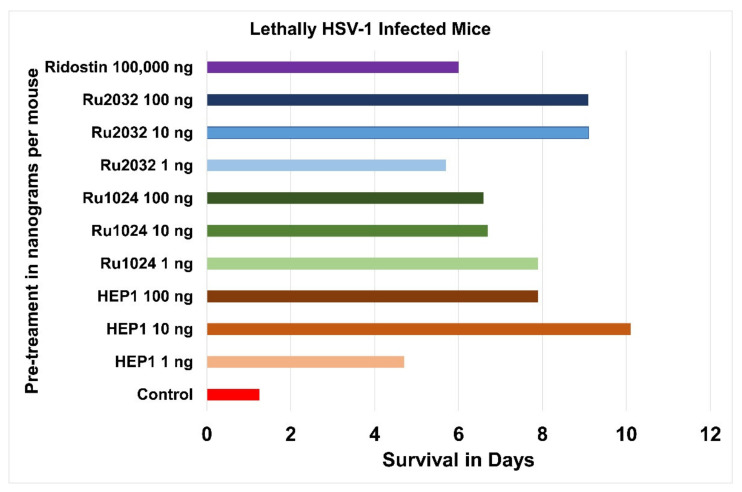
Adjuvant activity of ezrin hep-receptor peptides at three different concentrations (1 or 10 or 100 ng per mouse) conferring protection in the mouse lethal herpes virus infection model measured by the increase in survival in days. Red bar: no adjuvant (negative control mice), purple bar: Ridostin adjuvant (positive control mice) and hundred-fold concentration ranges of synthetic peptides that are ezrin hep-receptor mimics; blue-tones Ru2032 KEELMLRLQDYEE, green-tones Ru1024 EREKQMMREKEEL and brown-tones HEP1 TEKKRRETVEREKE.

**Figure 15 ijms-22-11688-f015:**
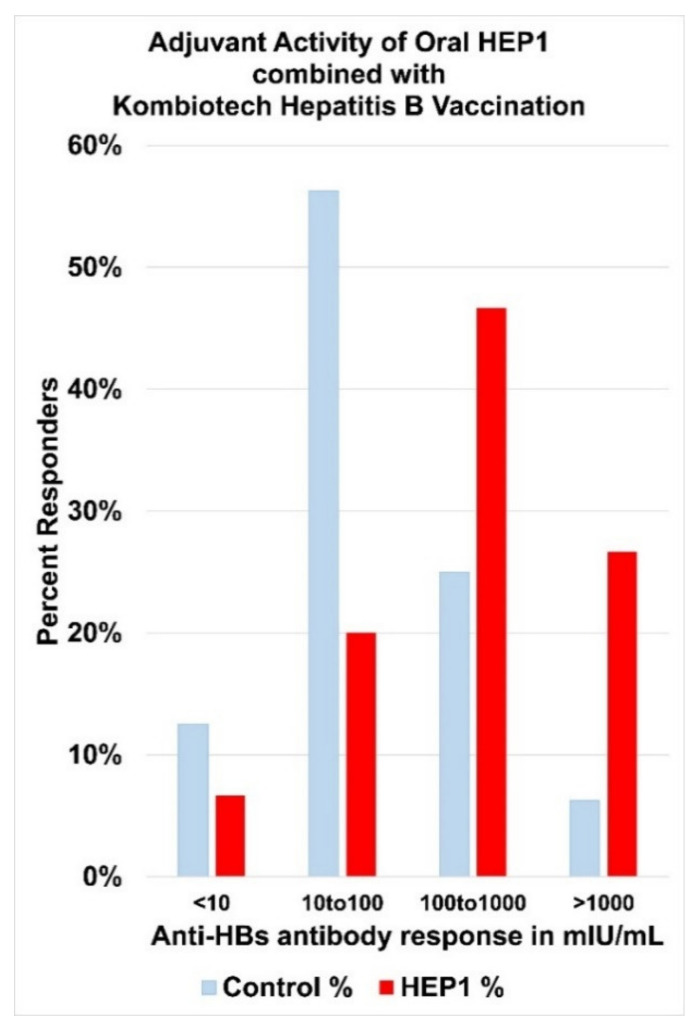
Adjuvant activity of oral 2 mg/2 mL of HEP1 solution given sub-lingually before each of the four Kombiotech hepatitis B Vaccination (KHBV) injections.

**Figure 16 ijms-22-11688-f016:**
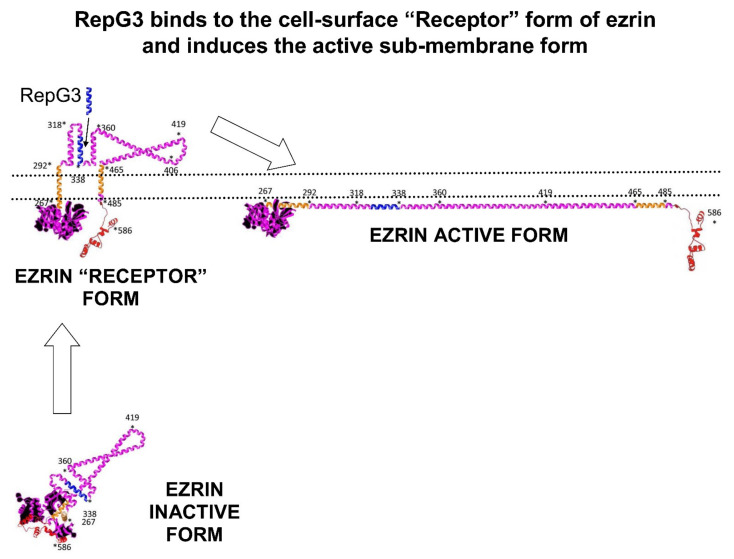
RepG3 may bind to a cell-surface “Receptor” form of ezrin, leading to allosteric conformation changes and the induction of the active sub-membrane form of ezrin followed by the assembly of protein signalling complexes. The white arrows represent the transitions from the inactive form of ezrin, to the “receptor” form of ezrin to the active form of ezrin.

**Figure 17 ijms-22-11688-f017:**
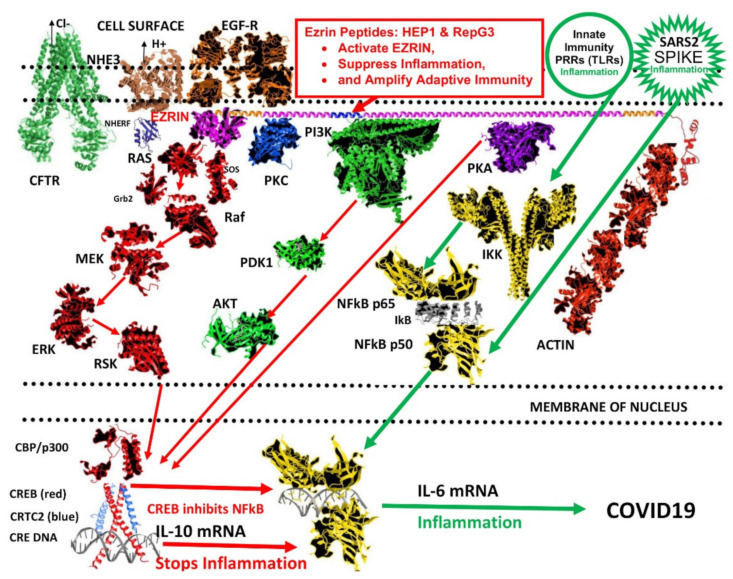
A hypothetical model of the cell-signalling pathways leading to the induction of excess inflammation by SARS-CoV-2, contrasted with the possible site of action of ezrin peptides at ezrin and different cell control pathways that lead to suppression of inflammation. (images produced in Adobe Photoshop and Microsoft Word). Red arrows represent cell signalling pathways leading to suppression of inflammation. Green arrow represent cell signalling pathways leading to activation of inflammation. SARS-CoV-2 spike-protein triggers cell-surface pattern recognition receptors (PRRs) such as toll-like receptors (TLRs) that are associated with innate immunity, to initiate inflammatory responses via multiple pathways to IKK (yellow) and activation of transcription factor NFkB (yellow), which migrates to the nucleus and binds to DNA at a spectrum of gene-promotors (grey), which initiate the expression of inflammatory cytokines such as IL-6. In contrast, active-ezrin assembles cell-signalling complexes associated with adaptive immunity such as PKA (purple), PI3K > PDK1 > AKT (green), PKC (blue) and SOS + Grb2 + RAS > Raf > MEK > ERK > RSK (red left), together with actin reorganization associated with receptor clustering and the formation of the immunological synapse (red right). In the nucleus, downstream effectors of PKA (purple) AKT (green) and RSK (red) activate transcription factor CREB (red) which scavenges CBP/p300 (red), so this co-factor is unavailable for NFkB mediated gene transcription (yellow). CREB (red) combines with CRTC2 (light blue) to engage CRE DNA and initiates the expression of anti-inflammatory IL-10, which also inhibits inflammation. It is interesting that cystic fibrosis symptoms are similar to COVID and the cystic fibrosis transmembrane conductance regulator CTFR (light green left) that controls mucus membrane hydration in the airways, is closely associated with ezrin via a complex with NHERF and NHE3. Ezrin activation is also controlled by the epidermal growth factor receptor EGF-R (brown). Signals induced by ezrin peptides, are thought to bias cell-signalling systems and transcription-factor usage, from innate immune responses to adaptive immune responses, suppressing inflammation in the process. (other ezrin-associated cell-surface proteins such as: CD44, CD43, ICAM1, ICAM2 and CD95 are not depicted to avoid cluttering the diagram).

## Data Availability

Not applicable.

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
