# Peer review of "Ezrin Peptide Therapy from HIV to COVID: Inhibition of Inflammation and Amplification of Adaptive Anti-Viral Immunity"

_ijms, 2021, doi:10.3390/ijms222111688_

Round 1

Reviewer 1 Report

The review article by Holmes and Ataullakhanov adresses an interesting topic that is hardly coved in the scientific literature. Ezrin peptides show interesting properties and potential applications for modulating immun response. The mechanism of action is clearly stated as hypothetic.
I would like to recommend this review article for publication, once some issues have been addressed.
1) There are several crytsal structures available that should be mentioned and a short paragraph about the 3D structure of ezrin proteins should be added.
2) The figures are not publication-ready. They are of poor quality, some of them look like scanned and copied (e.g. figure 3 and others). Some still contain labels in cyrillic letters, which suggests that they come from a different source, which is then not cited... Furthermore, the figure captions could be extended to inform the reader about the figure content and its implications.

Author Response

Thank you for your interest in our work.

  1. Unfortunately there is no X-ray crystal structure for the whole of human ezrin protein because the alpha domain is a flexible structure that does not form ordered crystals. Only the N-terminal FERM domain has been crystallised.
  2. I have revised the figures, replaced Cyrillic with Latin characters, and added headings. Unfortunately, the printing of the original Russian publications was not high quality. The origin of the figures in cited in the associated text.

Reviewer 2 Report

The article by Rupert D Holms and Ravshan I Ataullakhanov is a review focusing on the potential of human ezrin peptides to contribute to both inhibition of inflammation and anti-viral immunity.

Given the pandemic context and the potential of Ezrin Peptide to be used in the treatment of COVID, this review is timely. Nonetheless, I strongly advise authors to thoroughly check the overall writing of their manuscript, in particular I would recommend avoiding repetitions of descriptions and properties (some paragraphs seem to be very similar).

* Line 115: “the phosphorylation of Tyrosine-353 located in the ezrin Hep-receptor”. From Figure 2, when I check for the location of the corresponding Tyrosine, I see that it is number 353. I suggest checking this (small) mismatch in the numbering.

* Line 122 and following: “Of the 135 AVRI patients, 50% was due to Influenza, 13% was due to parainfluenza, 14% was due to adenovirus, 8% was due to RSV, 10% was due to mixed virus and 7% was due to unidentified virus.” → the addition of the percentage amounts for a total of 102% !!!

* Line 433 and following: “HEP1 is a 14 amino acid synthetic peptide mimicking part of the Alpha domain of human ezrin known as the Hep-receptor, which is formed by amino acid 324 and amino acid 337 inclusive. The Hep-receptor is a zip-like structure comprising of two anti-parallel alpha-helices carrying alternating positively and negatively charged side chains. The biological activity of fragments of HEP1 (see Figure 2) were investigated to determine the active regions of the peptide.”

I can’t figure out the reason why this structural description is placed here in the text ...

* In my opinion, the figures and their captions need further improvement and checking (see below).

Figures

All the proposed figures should have a comprehensive legend. Indeed, any figure should make sense by itself, without searching its description in the text. Especially in a review work.

Figure 1, 2, 13 and 17 don’t even have a title.

* Figure 3 → the titles of the y-axes are not written in English. I understand the figure is extracted from a previous article, but it seems reasonable to change things like the title if there are for instance in Russian so that any reader can understand the figure on first reading. In addition, the top panel of this figure uses statistical indicators (p) but there is no specification on how these quantities were evaluated.

* Figure 4 → the titles of the x and y-axes have to be written in English.

* Figure 5 → the title of the y-axis has to be written in English

* Figure 6 → the legend states that this figure illustrates the “Anti-inflammatory effects of Ezrin Peptides in vivo and in vitro”. I could be wrong, but I don’t see any in vitro results in this figure, and the title above the figure says “Ezrin Peptides Inhibition of IL6 Expression RepG3[V2] vs Gepon[HEP1] in vivo in Mouse DSS IBD model”. I advise you to thoroughly check the consistency between the text and the figures.

* Figure 7 → the points corresponding to the experimental measurements are not really visible. I suggest increasing their size, if possible.

* Figure 10 →I suggest to indicate the sequence of the different peptides in the figure (as it is done on Figure 11).

* Figure 11 and 12 → these two figures should be assembled in the same figure containing 2 panels. In its current form, I think there is an inversion of legend between Figure 11 and Figure 12. Indeed, the title of Figure 11 is “Two separate experiments confirmed the complex dose-response of the HEP1 adjuvant effect” while the two separate experiments Expt 1 abd Expt 2 are shown on Figure 12. Similarly, the title of Figure 12 is “The two separate peaks of adjuvant activity of HEP1 and ezrin peptide ...” while the Figure showing the results related to different peptides is Figure 11.

There is no reference or description of Figure 11 and 12 in the main text. This should be fixed.

* Figure 14 → I do not understand why the legend reads “Ezrin Peptide Adjuvant effect in Kombiotech (KHBV) Vaccination against Hepatitis B, in a Human Clinical Trial.” whereas the title of the figure as well as the title of the y-axis indicate that the experiments were carried out on mice.

* Figure 16 → I can not figure what is the take home message from this illustration and how it relates to the data presented before. I think the authors should extend its description and how it is connected to the other data.

* Figure 17 → I make the same remark for this figure as the one I just made above for Figure 16. I guess this illustration should be the one allowing the reader to get an overview of the potential of Ezrin Peptide Therapy and its probable mechanisms of action, but the lack of description (no title and no legend) and the very very short discussion make it very difficult to easily pinpoint the authors’ demonstration….

Minor corrections

* line 213 → “Clinical Symptoms (Days)Gepon Group n=50 (dark)”,

a space is missing: “ Clinical Symptoms (Days) Gepon Group n=50 (dark)”

* line 447 → “and other ezrin peptide fragments from 10 nanograms to 10 micrograms per mouse),”

there is a parenthesis missing: “and other ezrin peptide fragments (from 10 nanograms to 10 micrograms per mouse),”

Acronyms

* DSS should be defined on its first appearance (line 97) and not in line 355.

* line 106: “Ezrin and its homologues radixin and moesin (the ERM proteins)” → ERM has already been defined in line 64.

* RSV should be defined in line 124.

* SLTB should be defined on its first appearance (line 244) and not in line 330. I wonder if this acronym refers to the same thing as LTBS. If so, I suggest authors use only one of the acronyms throughout the text.

* LPS, used in Figure 6 is never defined in the text of the manuscript.

* SE should be defined on its first appearance (line 384) and not in line 394.

* HSV should be defined on its first appearance (line 481) and not in line 497

References

In my opinion, the use of latin numeration is not very easy for the reader because it is not a common practice.

Author Response

Thank you for your interest in our work and your detailed comments.

1. “Repetition” checked for repetitions.

Line 115. There is a one amino acid error in the literature on ezrin, between what is often referred to as Tyrosine-353, and the sequence numbering provided by Uniprot database.

Line 122. This is a number-rounding error which has been corrected.

Line 433 and on. Text moved to Line 98 to introduce Figure 2: Spectrum of Ezrin HEP-Receptor Peptides Investigated.

2. “Figures”

Titles

Figure 1 “Homology of the C-terminus of HIV gp120 and the Alpha domain of human ezrin”

Figure 2 “Ezrin peptides investigated that mimic the Hep-receptor in the Alpha domain of human ezrin”

Figure 13 “Over-lapping ezrin peptides mimicking the Hep-receptor of human ezrin”

Figure 17 “A hypothetical model of the cell-signalling pathways leading to the induction of excess inflammation by SARS-CoV-2 and the possible site of action of ezrin peptides and cell signalling induced by ezrin peptides that leads to suppression of inflammation .

English

Figure 3 Cyrillic replaced with English

Figure 4 Cyrillic replaced with English

Figure 5 Cyrillic replaced with English

Consistency

Figure 6 “Anti-inflammatory effects of ezrin peptides in the mouse dextran sulphate solution (DSS) model” [LPS: lipopolysaccharide]

Figure 7 visibility. Data points at breaks in slope

Figure 10 “Ezrin Peptide Adjuvant Activity of HEP1 and related peptides, enhances the number of IgM antibody-forming cells in the spleen in response to sheep erythrocyte primary immunisation. HP1-9 TEKKRRETV, HEP1 TEKKRRETVEREKE, HP6-14 RETVEREKE, HP4-14 KRRETVEREKE and HP-3-12 KKRRETVERE.”

Figures 11 + Figure 12 formatting error

A reference to figures 11 &12 added to end of section 6 text.

Figure 14 formatting error. New title “Adjuvant activity of ezrin Hep-receptor peptides at three different concentrations (1 or 10 or 100 nanograms per mouse) in the mouse lethal herpes virus infection model”

Figure 16 clarification

Figure 17 “A hypothetical model of the cell-signalling pathways leading to the induction of excess inflammation by SARS-CoV-2 and the possible site of action of ezrin peptides and cell signalling induced by ezrin peptides, which leads to suppres-sion of inflammation”.

Minor typos

Line 213 OK

Line 447 OK

Acronyms

Line 97 define DSS (not in line 355) OK

Line 106 repetition OK

Line 124 RSV? OK respiratory syncytial virus (RSV)

Line 244 SLTB? Not at 330 does SLTB = LTBS? OK

LPS used in figure 6 defined

Line 384 SE? (Not at 394) OK

Line 481 HSV? (not at 497) OK

References

General formatting error. I have replaced the Roman Numerals with Numbering for the end-note references.

Reviewer 3 Report

Dear Authors

the review is interesting nowadays, but it is not suitable for publication journal; my suggestion is to change the organization of the topic according to journal guidelines and aim and scope; my recommendation is major revision.

Author Response

Regarding your comment: Change the organisation of the topic according to journal guidelines and aim and scope: a major revision.

Please could you provide a more specific and detailed recommendation.

We have been working on the biological activity of synthetic peptides mimicking human ezrin for over twenty five years and there are a large number of publications relating to the clinical benefits of ezrin peptides in the Russian language. We are introducing this field of research to English speakers and our narrative is to explain why we started this work (the unexpected homology between HIV gp120 and human ezrin), the discovery of a board range of useful therapeutic responses to ezrin peptides in patients, the novel mechanism of action (anti-inflammatory combined with adaptive immune response amplification), and the recent discovery that ezrin peptides could be effective treatment for mild to moderate COVID. In future, we shall publish new results from formal clinical research on ezrin peptides as COVID therapy. However, we would like to alert our peers to investigate this field of research, when there is such an urgent need to find practical solution to the SARS-CoV-2 / COVID pandemic.

Round 2

Reviewer 2 Report

Overall, the authors took into account the comments that were made and broughht modifications accordingly. Nonetheless, I would like to insist on the following points :

* line 122: "the phosphorylation of Tyrosine-353 located in the ezrin Hep-receptor"

From Figure 2, when I check for the location of the corresponding Tyrosine, I see that it is number 354 (Sorry I did not write the good number in my last comments). I suggest correcting this (small) mismatch in the numbering.

* I insist on the fact that captions need further improvement : they should be more descriptive.

* Figure 8 → the points corresponding to the experimental measurements are not really visible. I suggest increasing their size, if possible.

* A more substantiating discussion (based on the description of the last figure for instance) would make sense and improve the overall impact and interest of the article. 

Author Response

Dear REVIEWER

Thank you for your additional comments on our manuscript which I have tried to address.

1. I have corrected the mismatch in numbering to Tyrosine 353 which is the generally accepted numbering in the literature.

2. I have tried to follow your recommendation to expand the captions. Please consider the significant additions.

3. I have added more material to the Discussion and final caption.

Round 3

Reviewer 2 Report

The authors took into account the comments that were made and broughht modifications accordingly.